# 3D Reconstruction of Non-Rigid Plants and Sensor Data Fusion for Agriculture Phenotyping

**DOI:** 10.3390/s21124115

**Published:** 2021-06-15

**Authors:** Gustavo Scalabrini Sampaio, Leandro A. Silva, Maurício Marengoni

**Affiliations:** 1Graduate Program in Electrical Engineering and Computing, Mackenzie Presbyterian University, Rua da Consolação, 896, Prédio 30, Consolação, São Paulo 01302-907, Brazil; leandroaugusto.silva@mackenzie.br; 2Department of Computer Science, Federal University of Minas Gerais, Avenida Antônio Carlos, 6627, Prédio do ICEx, Pampulha, Belo Horizonte 31270-901, Brazil; mmarengoni@dcc.ufmg.br

**Keywords:** 3D reconstruction, computer vision, sensor data fusion, robotics, agriculture

## Abstract

Technology has been promoting a great transformation in farming. The introduction of robotics; the use of sensors in the field; and the advances in computer vision; allow new systems to be developed to assist processes, such as phenotyping, of crop’s life cycle monitoring. This work presents, which we believe to be the first time, a system capable of generating 3D models of non-rigid corn plants, which can be used as a tool in the phenotyping process. The system is composed by two modules: an terrestrial acquisition module and a processing module. The terrestrial acquisition module is composed by a robot, equipped with an RGB-D camera and three sets of temperature, humidity, and luminosity sensors, that collects data in the field. The processing module conducts the non-rigid 3D plants reconstruction and merges the sensor data into these models. The work presented here also shows a novel technique for background removal in depth images, as well as efficient techniques for processing these images and the sensor data. Experiments have shown that from the models generated and the data collected, plant structural measurements can be performed accurately and the plant’s environment can be mapped, allowing the plant’s health to be evaluated and providing greater crop efficiency.

## 1. Introduction

The increased demand for agricultural products, whether to feed the world’s population, to supply inputs to the industry or even to trade these products on the commodities market, has encouraged producers to invest in technologies to improve production efficiency. Seeding and harvesting machines are examples of technologies responsible for the agricultural production substantial growth. The most recent figures from the Food and Agriculture Organization of the United Nations, going back to 2019, shows that about 12 billion tons of products from the field are produced annually [1]. This scenario is favorable for research and development of technologies that help the producer; not only at the beginning and ending stages of cultivation, but also at its intermediate phases, helping to identify problems earlier so decisions can be made faster and more accurate [1,2].

The plant growth monitoring is essential to improve crop efficiency and minimize losses. One of the processes that helps this monitoring is phenotyping. The analysis of the plant phenotype, that is, its morphological and physiological characteristics coming from the species genetics and the environmental influence, can be used to evaluate the individual and collective health of plants. Some phenotypic characteristics are: the plant size; the leaves width; the leaves and fruits density; and the angulation between the stems and the leaves; among others. This process and other technologies, such as smart irrigation systems and real-time monitoring, allow us to follow the plant’s life cycle, identifying health problems early on; to identify holes in the plantation, due to non-birth or poor formation; and to use fertilizers, pesticides, and water in a rational way. This monitoring aims to increase the efficiency of production, and therefore its volume, and to reduce labor and farm input costs [2,3,4,5].

With the research advances in the three-dimensional (3D) reconstruction area, some work has been proposed to use 3D reconstruction of plants in phenotyping processes. Most of these works are limited to controlled environments, avoiding natural factors such as wind and rain that cause movements in plants. Other works, which use field data, make use of static images only. These restrictions are due to the difficulty of generating 3D models of objects with dynamic characteristics, limiting the results seen so far to rigid reconstructions. However, non-rigid reconstruction is important to acquire more information about the objects in the scene, as it allows the accumulation of data from different frames. Moreover, non-rigid reconstruction of plants can be a first step towards modeling the plantation in a complete and continuous way, allowing its global visualization [2,3,6,7,8].

This work is inserted in this context. Not to perform phenotyping or plant growth analysis, but to provide inputs for these processes. Collecting data in the field and processing them manually is often unfeasible, since the producer would have to travel a great distance, depending on the size of the plantation, at various stages of the plant’s life [2,8,9]. Therefore, the objective of this work is to present a system for field data collection and processing to generate accurate 3D models of non-rigid corn plants in an uncontrolled environment, which, we believe, is the first system to do this; and merge this model with sensor data. Specifically, the system was composed of two modules: the acquisition module and the processing module. The acquisition module has aerial and terrestrial subsystems; both subsystems collect data from the field for post-processing; this work will present the development and characteristics of the terrestrial subsystem, called the terrestrial acquisition module from now on. This module is composed of a robot equipped with three sets of temperature, humidity, and luminosity sensors and an RGB-D (Red, Blue, Green and Depth) camera, and is responsible for navigating inside the plantation and collecting the data. In turn, the processing module is responsible for performing non-rigid 3D reconstruction of the plants generating a 3D model with precise measurements, as well as merging the sensors data into these models. The proposed system macro view can be seen in Figure 1. Besides these important contributions, the present work implements a novel technique for background removal in depth images, based on the logarithmic function. Several 3D reconstruction techniques were analyzed, and those that proved to be efficient were selected to be part of the system, such as point cloud treatment, surface reconstruction, and model deformation. The system was developed to work with corn crops, but can be adjusted in the future to meet the needs of other crops.

To present the system developed and the results achieved, this work was organized as follows: Section 2 presents the related works, focusing on 3D reconstruction and similar works that perform 3D reconstruction of plants; Section 3 presents the materials, methods, and techniques used to build the modules that compose the system; Section 4 presents the results, by showing the outputs provided by some tests; Section 5 presents the discussion of the results and proposals for future improvements to the system; Section 6 presents the conclusions.

## 2. Related Works

### 2.1. Robotics in Agriculture

The introduction of large machinery in agriculture, mainly to perform the seeding and harvesting of crops, has contributed significantly to consolidating today’s high agricultural production rates. Nowadays, agriculture is beginning to experience the phase of robotization [10,11,12]. Robotics in agriculture enables and facilitates the functionalities of precision agriculture, characterized by the application of techniques aimed to improve the efficiency of agricultural cultivation and better use and allocation of resources, such as soil, pesticides, and water [13]. In this sense, tasks in agriculture tend to be increasingly more precise with the participation of robots, as well as more sustainable.

From mapping agricultural activities, it can be observed that most of them, especially in the academic context, can already be performed by robots. Traction [14,15], seeding [16,17,18,19], detection and mapping [8,9], monitoring and phenotyping [2,20,21,22,23], pest control [24,25,26,27,28] and harvesting [29,30,31,32] are some activities that have already been addressed in proposing robotic systems for agriculture in different crops [33]. Among these activities, detection and mapping; monitoring and phenotyping; and pest control play a prominent role. These activities are performed in the intermediate phase of the plant’s life cycle to follow its growth.

The detection and mapping of plants is an essential activity, as it enables the other activities to be carried out. This task can be performed by robots equipped with geolocation sensors (Global Positioning System—GPS), three-dimensional sensors (Light Detection Furthermore, Ranging (LiDAR) and RGB-D cameras) and Simultaneous Localization and Mapping (SLAM) techniques, as the system proposed by Weiss and Biber (2011) [9] or the system proposed by Potena et al. (2019) [8], which use data coming from a robot and a drone for crop mapping. While detection and mapping can be performed only once during the plant’s life cycle, monitoring, phenotyping and pest control activities, which are strongly related, happen periodically and must be performed by flexible systems from the standpoint of locomotion and data collection. Shafiekhani et al. (2017) [2], for example, proposes the Vinobot, a robot capable of collecting individualized plant data through a rod with temperature, humidity, and luminosity sensors and a robotic arm with an RGB-D camera for capturing depth images. The rod and the robotic arm allow data collection at different times in the plant’s life in a flexible way, adapting to the size of the plants. The works by Mueller-Sim et al. (2017) [21] and Bao et al. (2019) [22] also present robots with similar configurations, but with distinct sensor and flexibility for data collection. After collecting data in the field, in these works, post-processing takes place to organize and make the data available. In the case of systems that collect depth images, point clouds or 3D models can be used for structural analysis processes. The analysis of works and techniques on 3D reconstruction of plants, therefore, are important to print to the models generated the highest possible accuracy, ensuring reliability to the system.

### 2.2. 3D Plants Reconstruction

With the development and cost reduction of 3D sensors, 3D reconstruction research has come under the attention of researchers. Engineering [34,35,36], architecture [37,38] and medicine [39] are some areas benefited by this technology. The basic process of 3D reconstruction can be described in the following steps: data acquisition; point cloud preprocessing; point cloud fusion, when sequential depth images or images from different sources are used; object model generation or update; and surface reconstruction [40]. Three-dimensional reconstruction can be classified in two ways: rigid and non-rigid 3D reconstruction. Both are optimization problems, where one seeks to minimize the error during the merging of the point clouds. However, in rigid 3D reconstruction, the objects in the scene are static, while in non-rigid 3D reconstruction, these objects are dynamic and have some level of movement [40,41,42]. The Iterative Closest Point (ICP) algorithm, proposed by Besl and McKay (1992) [41], is already well established for rigid 3D reconstructions, and several other works have augmented and improved it [43,44,45]. As for non-rigid 3D reconstruction, there is still no single strongly consolidated technique, but some works [46,47,48,49,50,51,52], such as DynamicFusion [46] and Fusion4D [48], have proposed techniques for model deformation that are shown to be very efficient.

The 3D reconstruction systems for agriculture, especially those aimed at creating 3D models of plants, can be classified into those for indoor environments (controlled) and those for open environments, using images from the field. Naturally, indoor systems have higher accuracy and quality compared to outdoor systems, since the images can be collected in a standardized way and without the influence of external elements such as rain and wind. In these systems, the processing flow of the depth images follows the traditional and some techniques seem to be trending. To perform point cloud fusion, when the case, ICP [53] and Coherent Point Drift (CPD) [7] techniques were used; for the reconstruction of the model surface, triangulated mesh generation techniques [3,6,7,54] and Non-uniform Rational B-Spline (NURBS) representation were mostly used [53,55]. Open environment oriented systems, on the other hand, are recent and are in early stages [2,22,56]. They basically use single depth images to reconstruct the 3D models of the plants, only treating the point clouds, and aim to generate accurate models for taking structural measurements. However, these systems share the double difficulty of proposing a data collection system and a data processing system for future use. Both indoor and outdoor systems share two characteristics: they aim to provide inputs for phenotyping processes; and they perform rigid 3D reconstruction of plants. From the analysis of these works, it was identified the absence of non-rigid 3D reconstruction in agricultural systems, an important process to allow natural characteristics, such as wind, to be considered in the 3D reconstruction; this type of reconstruction is explored in this work.

## 3. Proposed System: Materials and Methods

Considering the objective of this work: to use images and field data to perform a non-rigid 3D reconstruction of the plants and provide inputs for the phenotyping process, the proposed system was structured to collect the data and then process them. Therefore, its development was divided into two modules, specifically designed for these activities. The terrestrial acquisition module focuses mainly on the hardware components of the system, such as the robot, the sensors, and the RGB-D camera. The processing module is composed of the software responsible for generating the 3D model of the plant and merging this model with the sensor data. The integration between these modules happens through bag files [57]. The terrestrial acquisition module stores the data collected by the sensors in a bag file and the RGB-D videos captured in another bag file; the processing module, in turn, reads these files and performs the necessary processing in order to generate the 3D model of the plant and represent the sensor data in this model through its colorization. Next, the components of each of the modules, the development architectures, and the details of the techniques involved in the depth images processing will be presented.

### 3.1. Robotic System for Terrestrial Data Acquisition

Integrating and managing various components of a robotic system is a complex task. The Robot Operating System (ROS) offers an architecture and features that facilitate the integration and control of the components of a robot. Thus, the terrestrial acquisition module was developed having as base architecture the ROS architecture [57,58]. From this definition, the module’s equipment and the nodes that compose the system could be defined and developed.

The main component of the module is the Jackal robot (unmanned ground vehicle), developed by Clearpath Robotics [59]. This robot has actuators for navigation, an on-board computer, GPS, LiDAR, Inertial Measurement Unit (IMU), and a 32-bit microcontroller for motor control, all integrated into a ROS. The ROS master runs on the robot’s on-board computer, which has a collection of native nodes and topics to read data and control its equipment. Through the ROS architecture, it is possible to integrate custom equipment into the robot; either by creating external nodes and establishing communication with the native nodes or by connecting equipment to the on-board computer’s Universal Serial Bus (USB) ports and creating internal nodes to control this equipment.

Carefully analyzing the problem of monitoring a corn plantation, some variables that are important to follow its growth were defined; as the size of the plant varies over time, these variables should be followed at different heights. Thus, a 1.5 m aluminum rod was installed in the robot, with supports for installing the sensors. The supports can be positioned at different heights, allowing more flexibility for data collection according to the life moment of the plantation. The variables chosen to monitor the plants were: temperature, humidity, and luminosity. Through these variables, it is possible to evaluate the environmental conditions in which the plantation is inserted, allowing the phenotyping process to define actions to solve problems in plant growth. Besides the sensor supports, the rod also allows the installation of an RGB-D camera, at different heights. Through the depth images, it is possible to reconstruct 3D models of the plants. Figure 2 shows the robot with the rod, the RGB-D camera installed and the sensor support in detail.

The choice of sensors and RGB-D camera took into account the accuracy of data collection, the equipment size and the price. The RGB-D camera chosen was a stereo depth camera, the Intel RealSense D435i [60]; this camera has sensors capable of capturing depth images in low light environments, being ideal for captures in closed plantation [60,61]. The camera was connected directly to the robot’s on-board computer via USB. For the temperature and humidity acquisition, the HDC1080 [62] sensor was used; this sensor has a resolution of 14 bits, providing accuracy of ±2% for relative humidity and ±0.2°C for temperature and has measurement time of approximately 6.50 ms for relative humidity and 6.35ms for temperature. For luminosity acquisition, the BH1750FVI [63] sensor was used, with 16 bit resolution, maximum precision of ±20% (lux) and measurement time of approximately 120ms. The data collected by the sensors are read, using the Inter-Integrated Circuit ($I^{2}C$) communication protocol, and processed by an Arduino MEGA 2560 [64]; and, then, transmitted to the requesting nodes via ROS service. Data can be collected by synchronizing the sensors measurement times with the desired video time, allowing numerous collection configurations. The proposed system was set up to collect two seconds of video and only one measurement from each sensor for each plant. Three sets of sensors were installed on the rod and all cabling for sensor data transmission was mounted with Unshielded Twisted Pair (UTP) cables. The Arduino is powered by and communicates serially (*rosserial*) with the robot’s on-board computer also via the USB port. The collected data is stored in bag files that are read by the processing module to generate the 3D model of the plant and merge this model with the sensors data.

### 3.2. Non-Rigid 3D Plant Reconstruction System

The terrestrial acquisition module, described previously, generates data that serves as input to the processing module. The goal of this processing is to perform the 3D reconstruction of the object present in the depth video, by means of techniques capable of processing non-rigid elements in the scene. The processing flow resembles that presented by Newcombe, Fox, and Seitz (2015) [46] and others that follow the same processing line. The system was developed in C++ with libraries for 3D object processing, among them: Intel RealSense SDK 2.0 [65], Point Cloud Library (PCL) [66], Computational Geometry Algorithms Library (CGAL) [67] and cpu_tsdf Library [68].

The first operation of the system is to read the bag files generated by the terrestrial acquisition module. From these files, the frames of the depth videos and the data recorded by the sensors are extracted. Then, from those depth images, the input point cloud is generated. This procedure involves the removal of the background, using a novel technique based on the logarithmic function; standardization of the point cloud, using the octree data structure; removal of outliers using the average distance between points; and smoothing the points position using the Bilateral Smoothing technique. With the point cloud presenting as few problems as possible the input surface is reconstructed using the Advancing Front technique, based on the Delaunay triangulation. The first processed frame generates the first model, which is deformed at each iteration with a new input frame. For each iteration, the model surface is deformed to align with the input surface, also accumulating data. This operation involves describing the objects keypoints with the Signature of Histograms of Orientations (SHOT) technique; matching the keypoints (kd-tree search); deforming the model, using the Smoothed Rotation Enhanced As-Rigid-As Possible (SR-ARAP) technique; merging the surfaces, by merging the volumes described by the Truncated Signed Distance Function (TSDF); and further processing the model, removing new outliers, smoothing the points, and reconstructing the surface isotropically. After processing all the frames (key frames), the system merges this model with the data collected by the sensors through colorization. The colorized 3D model is then exported as a Polygon File Format (PLY) file [69], which can be effectively used as input for the phenotyping process. Figure 3 shows the flow described in graphical form. Further details of the operations mentioned will be presented next.

#### 3.2.1. Background Removal

The point clouds generated from the depth frames carry a very large volume of data; for the standard resolution of the camera used (640 × 480 pixels) the generated point cloud has 307,200 points that mostly represent the scene background. The objective of removing these points is to reduce the volume of data processed by the other operations of the system and improve the model accuracy.

The images collected in the field have characteristics that make it difficult to segment, and therefore to remove the background, using only RGB (Red, Green and Blue) images; the main ones are the similarity of colors and shapes of the plants, even though they are at different distances. For this reason, only the collected depth data were used. To perform the background removal operation, a new technique was proposed, based on the logarithmic function, which proved to be more efficient than the linear background removal. This new technique uses a percentage cutoff factor to define the maximum depth value that will be kept in the point cloud. However, this value is found by relating this percentage and the minimum $d_{min}$ and maximum $d_{max}$ depth distances captured by the camera to the behavior of the logarithmic function.

The logarithmic function presents a behavior that can be efficiently used in positioning problems. Given the logarithmic function $f\left(x\right)=\log_{b}x$ and considering f(x) represented in the ordinates axis and *x* represented in the abscissa axis; if a comparison with the linear function is done, it can be noticed that the logarithmic function curve presents a smooth transition from its values in the abscissa axis to a large portion of the ordinates axis values. This behavior changes, at a certain point of the ordinate values, becoming more aggressive in the abscissa values. This relationship between ordinates and abscissa varies according to the base *b* of the function. For the logarithmic function curve behavior to be used in background removal, it is necessary to relate the desired cutoff factor and depth distance references to the properties of this function. Two properties of the logarithmic function are important to establish this relationship: the first is that $\log_{b}1=0$; the second is that $\log_{b}b=1$. These properties can be used to convert percentages [0,1] to values ranging from [1,b], since $\left[\log_{b}1,\log_{b}b\right]$ corresponds to the values [0,1]. Thus, a background cutoff factor $f_{bg}$ can be found by the equation $f_{bg}=\log_{b}x_{b}$, where $x_{b}$ represents the cutoff distance in the logarithmic domain, found by linear conversion between [1,b] and $\left[d_{min},d_{max}\right]$. Therefore, to find a depth value given a cutoff factor, it is sufficient to perform the reverse path, i.e., the cutoff factor $f_{bg}$ must be converted by means of the exponential function to the interval [1,b] and linearly relate it to the interval $\left[d_{min},d_{max}\right]$. For this case, the equation $x_{b}=b^{f_{bg}}$ is sufficient to convert the cutoff factor $f_{bg}$ to the interval [1,b]; and the linear conversion between the intervals [1,b] and $\left[d_{min},d_{max}\right]$ can be done with the equation
(1)$$\frac{b-x_{b}}{b-1}=\frac{d_{max}-d_{bg}}{d_{max}-d_{min}}\;.$$
Developing this equation and considering $x_{b}=b^{f_{bg}}$, the absolute depth distance value $d_{bg}$, which corresponds to the point cloud background, is found with the equation
(2)$$d_{bg}=-\left(\frac{b-b^{f_{bg}}}{b-1}\right)\cdot \Delta d+d_{max}\;;$$
where $\Delta d=d_{max}-d_{min}$. Any depth value greater than $d_{bg}$ is removed from the input point cloud.

In practice, the proposed technique is a pass-through filter; however, the value that defines what is background or not is found using the logarithmic function. By the characteristic of this function, objects close to the camera can be prioritized, since it is possible to control the curve of the function by the logarithmic base. Furthermore, the proposed technique is less sensitive to the variation of minimum and maximum camera depths than the linear form, and remains effective for many scenarios of this variation. To aid the understanding of the proposed technique the Figure 4 and Figure 5 illustrate the theoretical removal of the background and a practical example of this removal. In Figure 4, it is possible to observe the smoother variation of the proposed technique; the linear removal using a cutoff factor of 50%, always cuts the background in the middle between the minimum and maximum distances captured by the camera, whereas the proposed technique varies closer to the minimum distance. This is the desired behavior, since it was considered that the main object of the scene is closest to the camera. In Figure 5, it is possible to see the variation of the background removal using different logarithmic bases. After the initial adjustment of the cutoff factor, the proposed technique is able to remove the background dynamically and efficiently.

#### 3.2.2. Point Cloud Standardization

The point cloud standardization process was also implemented to enable the control of the cloud data volume. This standardization allows adjusting the cloud resolution in a structured way, allowing to modify its accuracy according to the desired object in the scene. For the database studied, objects with large proportions can be described with a lower resolution than objects with smaller proportions without loss in model quality; i.e., at the beginning of the plant’s life cycle, the point clouds should present a higher resolution for model generation, whereas for adult plants this resolution can be lower.

To perform this operation, the octree data structure was used to adaptively group sets of cloud points into voxels, according to a predefined resolution. For each of the voxels created, a single point describes the group [66]. This data consolidation can happen in four different ways: centroid of the points belonging to the voxel; centroid of the voxel position; first point inserted into the voxel; and last point inserted into the voxel. The first two ways are well known, but the last two are new and can be used to prioritize points in a cloud, in operations involving data from different sources, such as multiple cameras. In the tests, the proposed system used the centroid of the points belonging to the voxel to describe it. Figure 6 presents two examples of standardization, with 3mm and 5mm of resolution.

#### 3.2.3. Outliers Removal

Removing the background and standardizing the point cloud are operations that allow the control of data volume to process. However, either due to noise or the presence of unwanted elements in the scene, such as small pieces of leaves from other plants, scattered points may be present in the depth data. These points are undesirable, since their presence can compromise the accuracy of the model surface reconstruction and negatively influence the fusion operation of the input and model surfaces. For this reason, the outlier removal operation was implemented in the system.

This operation aims to remove points that have a distance among their neighbors outside the cloud pattern. To find these points, the average distance between points in the cloud must be calculated. Given a point cloud with *M* points and setting *N* neighbors, the average distance among points in the cloud $d_{m}$ can be calculated [70] by the equation
(3)$$d_{m}=\frac{1}{M}\sum_{i=1}^{M}\left(\frac{1}{N}\sum_{j=1}^{N}\sqrt{{\left(x_{i}-x_{j}\right)}^{2}+{\left(y_{i}-y_{j}\right)}^{2}+{\left(z_{i}-z_{j}\right)}^{2}}\right)\;.$$
Then, for each point in the cloud, the distance between this point and its neighbors $d_{local}$ is calculated and compared to the average distance $d_{m}$ of the points in the cloud times a factor $f_{out}$ (parameter), for example, twice the average distance ($f_{out}=2$). If $d_{local}>d_{m}\cdot f_{out}$ the analyzed point is considered an outlier and removed from the point cloud. Figure 7 presets an example of outliers removal for $f_{out}=2$, where circles indicate regions of outliers.

#### 3.2.4. Point Cloud Smoothing

Similarly to the outlier removal operation, the smoothing of the cloud points position was implemented to improve the surface reconstruction quality. When collecting depth data, the camera may add small ripples to the description of originally flat regions, a factor caused by inaccuracies in the depth data capture. This feature contributes to a decrease in the surface reconstruction quality and description of keypoints in the surface fusion operation. The smoothing of the cloud points corrects these imperfections, providing greater accuracy in subsequent operations and in the generated model. To perform this operation the Bilateral Smoothing technique was used, composed of two parts: smoothing normals; and points repositioning based on the adjusted normals.

The Bilateral Smoothing technique needs the normal vectors to the implicit surface of the cloud points and its neighbors. These vectors are fitted, minimizing the distance to their neighbors, to describe smooth surfaces. The points are then positioned to fit the normals, so that they also form smooth surfaces. The estimation of the normal vector for each point is performed using the Principal Component Analysis (PCA) technique, which uses the eigenvalues and eigenvectors of the point covariance matrix. The smallest eigenvectors correspond to the best normal vectors with respect to the plane defined by the neighboring points, and the vector of this group with the smallest eigenvalue has the smallest direction variation and is perpendicular to the directions of greatest variation, corresponding to the normal vector to the surface [66].

The normals are iteratively adjusted, taking as a parameter the difference in distance from neighboring normals. The goal is to minimize these distances so that the discontinuities remain in separate groups, making continuous surfaces uniform and keeping the edges of the object [71]. After updating the normals, the cloud points are repositioned using the Locally Optimal Projector (LOP) technique proposed by Lipman et al. (2007) [72] and modified by Huang et al. (2013) [71] for the context of the Bilateral Smoothing technique. LOP redistributes a set of points in a way that adheres to the implicit shape described by those points. The smoothing operation can be done iteratively, allowing the control of the smoothing level. Figure 8 presents three examples of point cloud smoothing with different iteration numbers.

#### 3.2.5. Advancing Front Surface Reconstruction

There are two forms of surface description of the objects processed by the proposed system: triangulated mesh (explicit representation), used to generate the deformation graph and the model; and TSDF volume (implicit representation), used to perform the fusion between the input surface and the model. The technique used to reconstruct the triangulated mesh was the Advancing Front [73,74]. This technique is based on the Delaunay triangulation, but has higher performance in terms of accuracy and quality, since the best triangles are formed sequentially. Another advantage of this technique is the ability to fill holes without having to perform this operation as preprocessing.

In the Advancing Front algorithm the initial triangle for mesh processing is defined by the size of the radius of the Delaunay triangles, defined by the circle passing over the three vertices of the triangle; the triangle with the smallest radius serves as the starting point of the algorithm. Then, the algorithm creates and maintains a list of candidate triangles to integrate the reconstructed surface. The list update happens every time a new triangle is incorporated into the partial surface, since new edges are made available for the reconstruction to move forward. For this list formation, there are four possibilities for a triangle to be a candidate to compose the surface: by extension; hole filling; ear filling; and glueing [73,74].

From the list of candidates, the triangle that has the highest plausibility degree is incorporated into the partial surface. The algorithm performs the update and definition of new triangles until all edges are verified. If a discontinuity occurs, the system performs the radius verification of the Delaunay triangles again, starting from the triangle with the smallest radius [73,74]. Figure 9 presents an example of surface reconstruction using the Advancing Front technique.

#### 3.2.6. Keypoints Definition, Description and Correspondence

The model deformation operation, for the model and input surfaces fusion, needs the new position data for the vertices considered key, or keypoints. For this, implementations were necessary to define these vertices, describe them and find the pairs of similar vertices between the surfaces. This whole process, involving the keypoints, happens by means of a deformation graph. The generation of this graph happens by refining the object’s surface, defining a new triangulated structure of lower resolution, but which faithfully describes the original surface. In the proposed system, the Incremental Triangle-based Isotropic Remeshing Algorithm [75] was implemented to generate the deformation graph. This algorithm performs simple operations incrementally on the surface edges and vertices; based on a predefined edge size, splitting, collapsing, inversion, as well as Laplacian smoothing (tangential projection) operations are performed. The operations are performed to describe a graph with triangles close to equilateral triangles; the Laplacian smoothing operation, in turn, seeks to adjust the triangles so the surface of the deformation graph is as close as possible to the original surface.

All vertices of the deformation graph mesh are considered keypoints. This process is performed for the input and model surfaces; and a mapping process is performed to maintain a relationship between the keypoints of the deformation graphs and the vertices of the source surface. With the keypoints defined the system is able to process the description and correspondence between these vertices.

In order for the model deformation operation to take place properly, the system must define the target position of the keypoints of the model deformation graph as best as possible. This definition is possible by matching the features of the model’s keypoints and the input keypoints. Features such as position, topology, and neighborhood can be used to quantitatively describe a vertex of a triangulated mesh. Subsequently, these features can be compared, establishing a relationship of closeness and correspondence. By finding these correspondences, the target position of the model’s deformation graph keypoints can be defined as the positions of their pairs in the input surface deformation graph. The description of the keypoints was implemented using the Signature of Histograms of Orientations (SHOT) technique [76].

Traditional 3D descriptors can be divided into two groups: descriptors by signature and descriptors by histograms. The first ones describe vertices by analyzing their neighborhood (support), defining an invariant reference frame, and encoding geometric measures of this support. The second ones describe the support by accumulating local geometric and topological measures, such as triangles areas and vertex counts in histograms. The SHOT descriptor brings these two forms of description together to create a stronger descriptor. With an isotropic spherical grid as the support structure, first-order entity histograms, such as the normals of the vertices in the support, and geometric information, such as the position of vertices, are extracted [76]. This support is sectioned along the radial, azimuth, and elevation axes; for each spatial division, a local histogram is computed. The construction of each local histogram takes into account the number of vertices and the value of the function that relates the angle between the normal of each vertex of the division and the normal of the keypoint.

To find the correspondence between the vertices of the deformation graphs, the search was implemented using the *k*-dimensional tree data structure (kd-tree). This structure is a specialization of the binary tree and allows for efficient search among nearest neighbors [66]. The *k* dimension used in the search is the SHOT descriptor of each vertex. The search process implemented in the proposed system was two-way search, improving the matching accuracy. In this process, first the vertices of the model deformation graph are compared with the vertices of the input deformation graph, then the input vertices are compared with those of the model; the matching pairs of vertices present in the two searches are defined as matching pairs. Figure 10 presents two examples of regions where keypoints are related, considering a sequence of depth images; the black lines indicate the match between vertices.

#### 3.2.7. Model Deformation

The keypoints definition, description, and matching between deformation graphs allow the matching points to form pairs of source and target positions for the model deformation process. Since the surface model is the one to be deformed, a proximity mapping was implemented between the vertices of this surface and the vertices of its deformation graph. Thus, the keypoints of the model surface are given target positions, making the model suitable to be deformed.

The technique implemented to perform the model deformation was the Smoothed Rotation Enhanced As-Rigid-As Possible (SR-ARAP) technique [77], a specialization of the As-Rigid-As Possible (ARAP) technique [78]. The deformation algorithm is similar to the ARAP deformation; however, a vertex rotation evaluation component is added to the energy function, providing greater transition smoothness in regions with steeper deformation. This technique is based on the Laplacian warping technique and uses a weighting scheme between mesh edges to drive the movement of vertices that are not keypoints.

In the deformation process, the surface keypoints are considered control vertices, since their movement influences the positioning of the other vertices, which compose the deformation region of interest. This movement propagation is accomplished by means of weights for the edges, which define what will be the movement portion of the vertex in relation to the moved control vertex. For the proposed system, the cotangent weighting scheme was implemented, which defines the weight by the ratio between the angles opposite to a given edge. Given the angles $\theta_{A}$ and $\theta_{B}$ opposite to an edge $e_{i}$, the weight of that edge $w_{e_{i}}$ is given by Equation [79]
(4)$$w_{e_{i}}=\cot\left(\theta_{A}\right)+\cot\left(\theta_{B}\right)\;.$$

Minimizing an energy function $E(S^{\prime })$ leads to an optimal deformation; which converges, with small error, the S surface to a deformed $S^{\prime }$ surface, considering the deformation constraints. The SR-ARAP technique introduces a bend element to the energy function; the addition of this term allows a control vertex and its neighborhood to obtain an optimal rotation in the deformation process. Thus, given a surface S with *n* vertices $v_{i}i\in \left\{1\cdots n\right\}$ and *k* control vertices, the energy function defining the SR-ARAP deformation is given by Equation [77,79]
(5)$$E\left(S^{\prime }\right)=\sum_{v_{i}\in S}\sum_{v_{j}\in N\left(v_{i}\right)}w_{ij}{∥\left(v_{i}^{\prime }-v_{j}^{\prime }\right)-R_{i}\left(v_{i}-v_{j}\right)∥}^{2}+\alpha A{∥R_{i}-R_{j}∥}_{F}^{2}\;;$$
$R_{i}$ is the optimal 3×3 rotation matrix from source vertex position to target position; $w_{ij}$ represents the edge weight; $N(v_{i})$ represents the set of vertices adjacent to $v_{i}$ in S; αA is the scalar weighting, where *A* represents the surface area to make the energy invariant to global scaling; *F* indicates that the difference between the rotation matrices of the vertices and their neighbors is calculated by the Frobenius matrix norm; and $N(v_{i}^{\prime })$ represents the new position of the vertices $N(v_{i})$ after deformation. The minimization of this equation can be simplified to the minimization of the energy of each vertex $v_{i}$, considering that there are two unknowns: the new positions of vertices that are not part of the set of control vertices; and the rotation matrices $R_{i}$. This minimization can be performed in two steps (local/global method) and iteratively, to further improve the deformation result [77,78,79]. In the proposed system, two stopping parameters were set for the deformation algorithm; either a pre-defined maximum number of iterations or a minimum value of the energy function $E(S^{\prime })$. The Figure 11 and Figure 12 present examples of surface deformation in two different regions of the plant, considering a sequence of depth images; in these figures it is possible to visualize how the proposed system performs the non-rigid deformations to align the surfaces for subsequent fusion.

#### 3.2.8. Input and Model Surfaces Fusion

The model deformation operation results in a surface aligned to the input surface of a given system iteration. However, these two surfaces have different areas; the model surface describes accumulated regions of the previously processed surfaces, while the input surface may contain new elements, arising from the object’s movement. Therefore, in order for the model to describe these new regions it is necessary to update it, accumulating these new parts of the objects to the existing parts. Another important factor in this surface fusion is that this process is able to remove small gaps between them, arising from errors in the deformation process, consolidating the model into a surface free of duplicate regions.

In the proposed system, the fusion of the input and model surfaces happens by updating the Truncated Signed Distance Function (TSDF) [80] volumetric representation of the model with respect to the same input representation. This implicit form of surface representation defines Signed Distance Function (SDF) values for the voxels of a uniform grid; where SDF values close to 0 represent the surface of the object. The SDF values can be truncated at ±t (TSDF), removing values far from the object’s surface and reducing the amount of data processed and memory usage. Through this representation, it is possible to perform volume fusion by weighted averaging of TSDF values [80,81].

For the surface fusion to be performed by means of TSDF volumes (implicit representation) it was necessary to implement conversion techniques between this form of representation and explicit representations (triangulated meshes). The conversion from explicit to implicit representation was performed by an adaptive grid using the octree data structure [68], where for each voxel, a TSDF value and a weight (uncertainty) were associated. To perform the conversion from the implicit representation to the explicit representation, the marching cubes technique was used [66,82].

The surfaces fusion always happens at the end of each input frame processing. Figure 13 presents an example of surfaces fusion; in this figure, it is possible to see the consolidation of the surfaces and the disappearance of small spaces between them. In an iterative way, this process allows the creation of a precise model with data accumulation over time. After processing the last frame, the model vertices are smoothed, new outliers are removed, and a new isometric surface is regenerated, thus standardizing the generated model.

#### 3.2.9. Model and Sensors Data Fusion

After processing all the input frames and generating the plant model, the system performs the fusion of the data collected by the sensors with the model. This fusion happens through the colorization of the model regions, according to the sensor values of the heights where they were installed; in the case of the proposed system, three heights. This way of fusion allows a quick visualization of the environmental conditions in which the plant is inserted. The system implements the data fusion in a separate way; that is, *n* copies of the models are created, one for each variable collected. To perform the model colorization, the linear interpolation of the RGB color space channels was implemented, relating the sensor heights; the sensor values; and parametric values of limits, of the variables and their colors. It was considered the linear interpolation Equation [83]
(6)$$c\left(x\right)=c_{min}+\frac{x-x_{min}}{x_{max}-x_{min}}\cdot \left(c_{max}-c_{min}\right)\;;$$
where c(x) represents the value of a given channel of the RGB color space, for a reference value *x*; $c_{min}$ and $c_{max}$, the limit values of the color channel; $x_{min}$ and $x_{max}$, the limit values of the reference value *x*.

Initially, the system defines the $RGB_{i}$ values for each measurement of the $s_{i}$ sensors, by means of Equation (Equation 6); taking into account the measurement and color limits for each variable (parameters). Thus, for each of the heights $h_{i}$, a $RGB_{i}$ value is defined. The processing to define the color of each vertex on the surface model involves the values $h_{i}$, $RGB_{i}$ and the vertex position. The pairs $(h_{i-1}$, $h_{i})$ and $(RGB_{i-1}$, $RGB_{i})$ replace, in Equation (Equation 6), $\left(x_{min},x_{max}\right)$ and $\left(c_{min},c_{max}\right)$, respectively; and the reference value *x* is replaced by the *y* axis value of the vertex position. As a result, the model surface is gradient colored, with the sensor heights as stop points. To illustrate this data fusion, Figure 14 presents an example of model and temperature sensor data fusion. The next section presents some outputs of the proposed system as results, from model generation to sensor data fusion.

## 4. Results

During the system development, several minor tests were conducted to verify the implemented functionalities and evaluate possible adjustments. Although these tests used images from plants belonging to the target database, new tests were conducted with the fully processing model for the 3D reconstruction implemented. From these tests, it was possible to collect data in the field and produce system outputs in a standardized way, which will be presented next.

First, the terrestrial acquisition module was configured and the camera was calibrated; then the camera and sensors were positioned on the rod. The camera was positioned at 1m height from the ground without tilting, while the sensors were positioned at 0.35m, 1.35m and 1.70m. After these procedures, the module was taken to the field, in a corn plantation, where the depth images and sensor data were collected. Thus, the depth camera was positioned about 40cm from the plants and the sensors about 5cm from the stems. For each plant, a two seconds video was recorded by the camera and one measurement from each sensor was collected. Data were collected both from inside and from outside of the plantation at two periods of the day; one collection sequence in the morning and one in the afternoon. In the morning the sky was completely cloudy, but in the afternoon the weather got better and the sky was clear with high incidence of sunlight on the plants. These weather conditions allowed to verify differences in data collection, especially in sensor values.

Before proceeding with data processing, a validation was performed on the collected depth data. Thus, it was possible to verify that the data used to generate the models were accurate and reliable, since these measurements can be used in phenotyping processes. Thus, static depth images were collected from 20 plants, so that the whole plant was captured (camera at 3m distance), then plant heights extracted from these images were compared with the heights measured manually. Figure 15 presents the scatter plot of the measurements. It can be seen that the scatter was low, resulting in a Root Mean Square Error (RMSE) of 0.022m or, if normalized by range (NRMSE), 1.75% error. These performance metrics indicate that the collected depth images have good accuracy and can be used for model generation.

The images and data were then processed. The system was set up to process two seconds of video, equivalent to 180 frames; the models, however, were generated using the sequence of frames from frame 30 to frame 150, with a 10 frame interval, totaling 13 frames processed. This initial frame cutting was necessary due to the camera startup time, which causes image distortions; the final frame cutting was done to maintain the time symmetry of the data used. Figure 16 presents some examples of generated models. In this figure, it is possible to see, besides the final model, the optical image, the depth image and the main processing steps. It can also be seen that, by means of these models, it is possible to extract measurements such as leaf width and length, angle between leaves and stems, measurements of the corn cobs, etc. To show the ability of the system to reconstruct the model in a non-rigid way and accumulate data over time, Figure 17 presents the models overlap of the processed frames, as well as the initial model and the final generated model. In Figure 17a,b, it is possible to visualize the movement of the plant leaves, especially in the overlapping detail section; even with this movement, the system performs the necessary deformations in the model to generate a final model without the presence of multiple leaves or undue deformations. Figure 17c presents a sharp example of data accumulation over time; in the initial frame the partial model presents thin leaves and even the absence of leaves, such as the lower right leaf; after the merging sequence, the final model presents thicker leaves and the presence of the lower right leaf.

Figure 18, in turn, exemplifies the fusion of sensor data with the models. This figure shows four different cases in data collection, so it was possible to observe all possible variations of the collected data. The models (a) and (b) were generated with data collected at the same time of day, which presented cloudy skies; it is possible to observe that plant (a) presents lower temperature, higher humidity, and progressive illumination in comparison with plant (b). For being inside the row, the plant (a) is inserted in an environment that varies more; its base, which is in the middle of other plants, receives less illumination and heat, but retains more humidity, in relation to the top of the plant. This behavior is different from the plant outside the row, inserted in a more uniform environment. The plats (c) and (d) present the same relationship as the plants (a) and (b); however, the data collection took place at a time with greater incidence of sunlight, increasing the temperature and luminosity and reducing the humidity.

## 5. Discussion

The results presented in the previous section show that the system is able to accurately produce 3D models of non-rigid corn plants from images and field data. This feature is essential for the development of crop inspection techniques, especially when considering future applications of systems that reconstruct the crop as a whole and not just individual plants. In applications of this size, inherent characteristics of open environments must be considered; one of them is the constant movement of the crop, due to wind or rain. The models generated can be used as inputs for phenotyping processes or structural measurements, while the fusion of sensor data to the model allows a quick visualization of the data collected in the field and a quick evaluation of the environment in which the plant is inserted.

Despite the goal achieved and the contribution of this work in the area, some characteristics of the system can still be further explored and new techniques or improvements in existing techniques can be studied. Starting with the processing form; the proposed system was not developed for real-time applications, since its functionalities were developed to be executed in the Central Process Unit (CPU) using threads. This processing could be implemented with parallelism in the Graphics Processing Unit (GPU), significantly increasing the processing speed. The system was design for phenotyping purposes, which usually uses small plantation areas. With the system developed for GPU processing, a larger area could be analyzed. From the point of view of the implemented techniques, some of them could be re-evaluated for performance improvement, and others could be replaced by new techniques in future work: analyze other methods for outlier removal, trying to eliminate the step of calculating the distances between points; analyze other surface reconstruction methods, with the aim of increasing the accuracy of the model; improve the correspondence index between the deformation graphs vertices, since this correspondence directly influences the deformation quality of the model; study the possibility of using a logarithmic weight scheme between the edges of the model, to increase the performance of the deformation; add new terms to the energy function, such as a possible logarithmic term for positioning between vertices; perform tests throughout the plant’s life cycle, not only in its adult phase; these are some examples of improvements for the proposed system.

Future works, using not only the concepts presented in this work, but consolidating other concepts from previous works, could be proposed for a broader exploration and inspection of the plantation: use of drones for data collection; use of mechanical arms for more dynamic collection of both data and images; fusion of aerial data (and images) and terrestrial data; new ways to merge data with 3D models, for example merging temperature and humidity data into the same color scheme; among others. The main motivation for the development of these future works is the importance that agricultural crops represent, from a food and goods production point of view. By improving the efficiency of inspection and monitoring of the plant’s life cycle, production in the field should increase.

## 6. Conclusions

Considering the current and future needs of agriculture, this work presented a system for field data acquisition and, which we believe to be for the first time, a non-rigid 3D reconstruction system for plants, to provide inputs for phenotyping processes. Through the developed modules, the field data collection and the model generation provide to the producer more efficiency in the plantation inspection and some inputs for plant structural evaluation.

The terrestrial acquisition module developed was composed of a robot, to which sensors and a depth camera were attached; thus, it was possible to collect data in the field in a more efficient way when compared to manual collection. The processing module was developed by implementing efficient techniques, evaluated from previous works, as well as new ones, such as the background removal based on the logarithmic function. With high depth data collection accuracy (error less than 1.75%), it was possible to generate models from depth video and merge these models with sensor data. Future improvements were presented in order to provide the system with higher processing speed, higher accuracy in the models generation, and alternative ways of data fusion.

The present work and other related works certainly contribute to the introduction of new technologies in the field, providing greater production volume and cost reduction. Not only that, they also encourage innovative works that also look at the field, since the resources produced there are of utmost importance for food and to provide inputs for the processing industry.

## Figures and Tables

**Figure 1 sensors-21-04115-f001:**
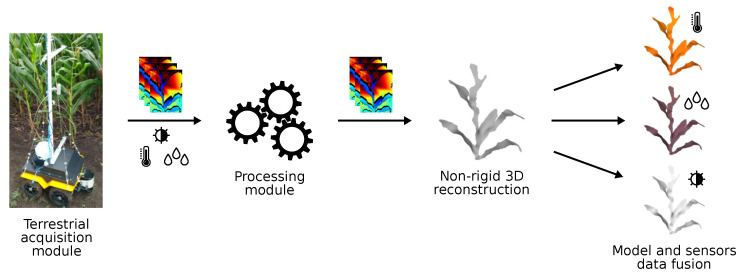
Proposed system macro view. The terrestrial acquisition module collects temperature, humidity, and luminosity data and RGB-D videos; the processing module performs a non-rigid 3D reconstruction of the plants and merge the sensors data into these models. With these modules, the system is able to generate inputs for phenotyping processes.

**Figure 2 sensors-21-04115-f002:**
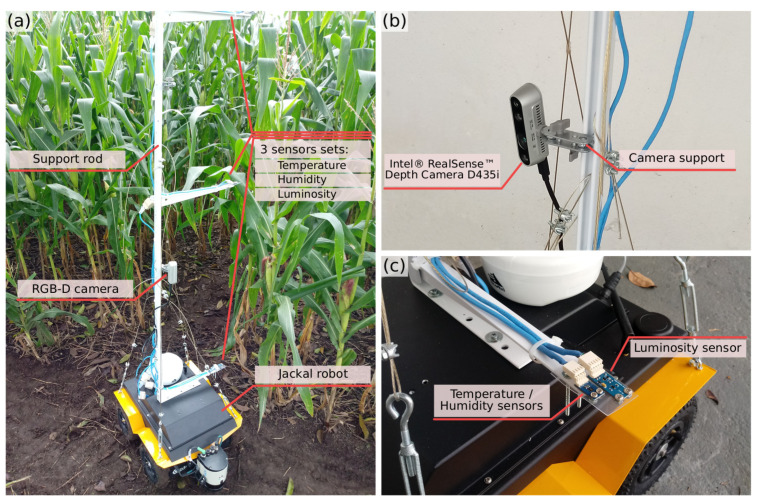
Proposed terrestrial acquisition module. (**a**) Overview; (**b**) RGB-D camera; and (**c**) sensor support.

**Figure 3 sensors-21-04115-f003:**
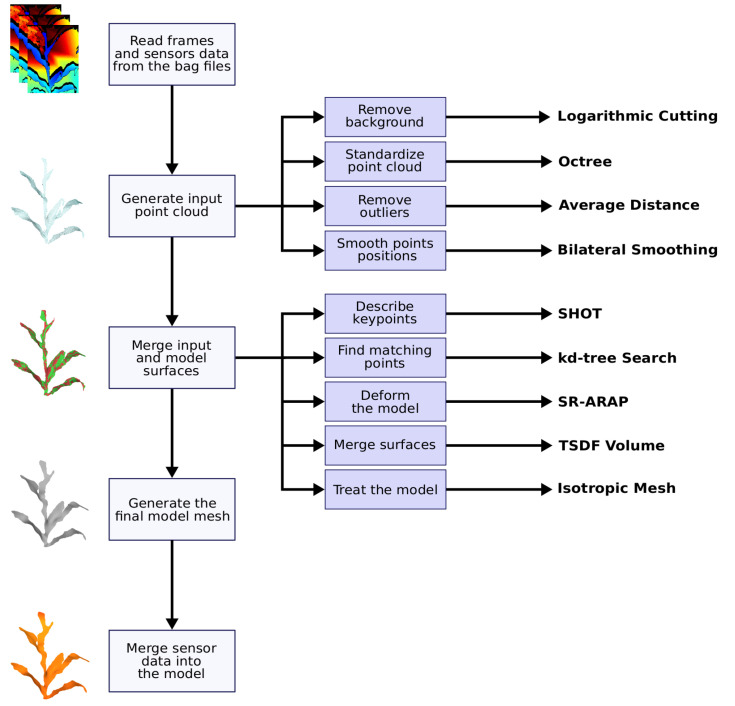
Proposed system processing flow.

**Figure 4 sensors-21-04115-f004:**
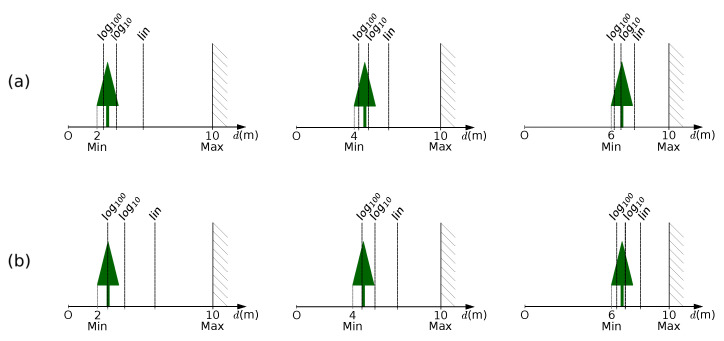
Examples of background removal for two logarithmic bases (10 and 100) of the proposed technique at different distances, compared to the linear form (lin). (**a**) Considering a cutoff factor of 0.4 and (**b**) considering a cutoff factor of 0.5.

**Figure 5 sensors-21-04115-f005:**
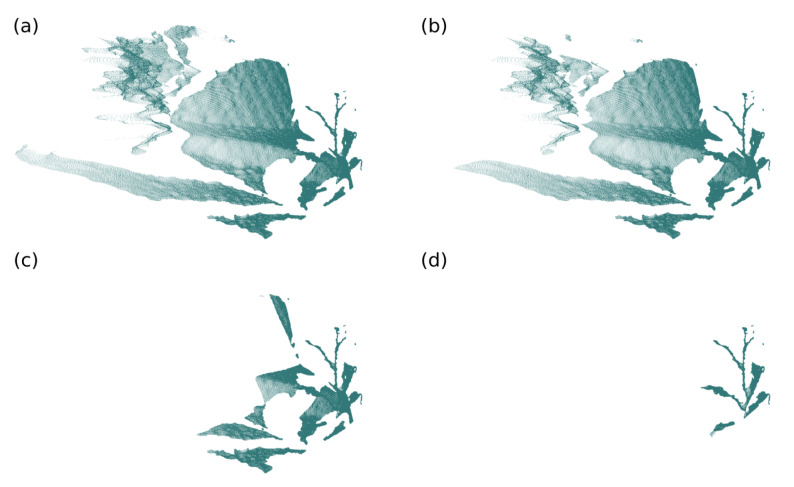
Example of background removal in a point cloud considering a cutoff factor of 0.2. (**a**) Input point cloud ($d_{min}=0.29\;m$ and $d_{max}=22.9\;m$); (**b**) linear cutting (at 4.81m); (**c**) cutting by the proposed technique with base 10 (at 1.76m); (**d**) cutting by the proposed technique with base 100 (at 0.63m).

**Figure 6 sensors-21-04115-f006:**
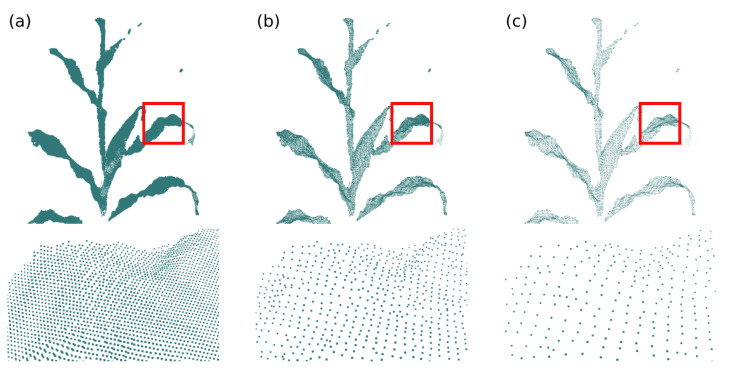
Point cloud standardization. (**a**) Input point cloud; (**b**) point cloud with 3mm resolution and (**c**) point cloud with 5mm resolution.

**Figure 7 sensors-21-04115-f007:**
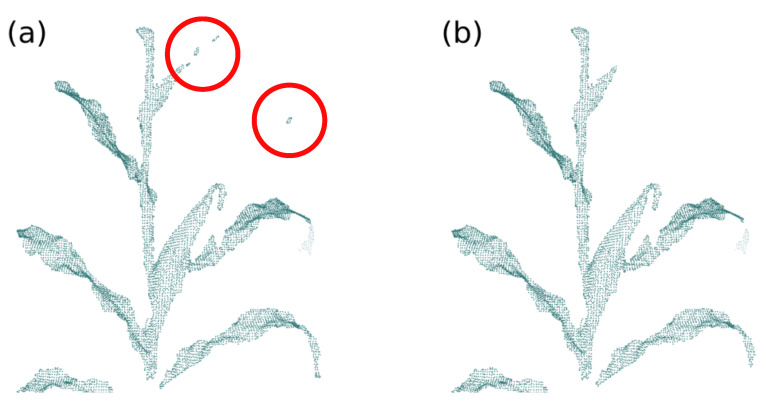
Outliers removal. (**a**) Input point cloud, the red circles indicate regions of outliers and (**b**) point cloud after the outliers removal operation ($f_{out}=2$).

**Figure 8 sensors-21-04115-f008:**
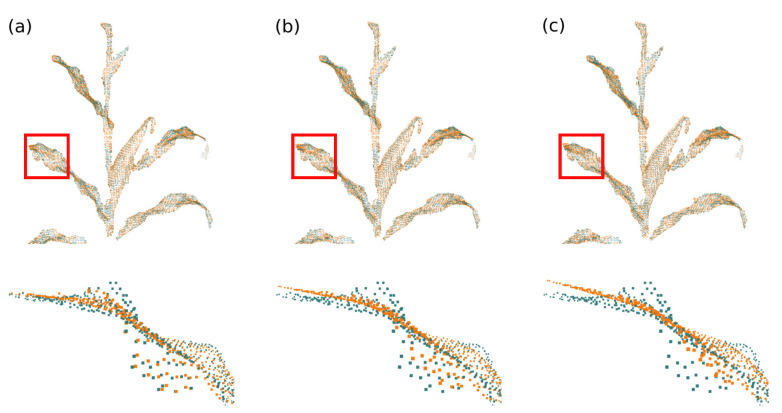
Point cloud smoothing. The orange point cloud indicates the smoothed point cloud, with (**a**) one iteration; (**b**) two iterations and (**c**) three iterations.

**Figure 9 sensors-21-04115-f009:**
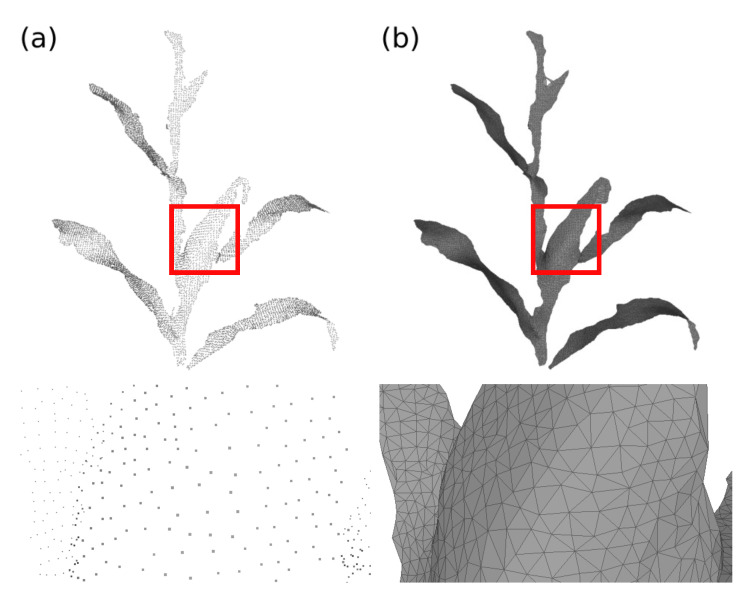
Advancing Front surface reconstruction. (**a**) Input point cloud and (**b**) surface reconstructed with the Advancing Front technique.

**Figure 10 sensors-21-04115-f010:**
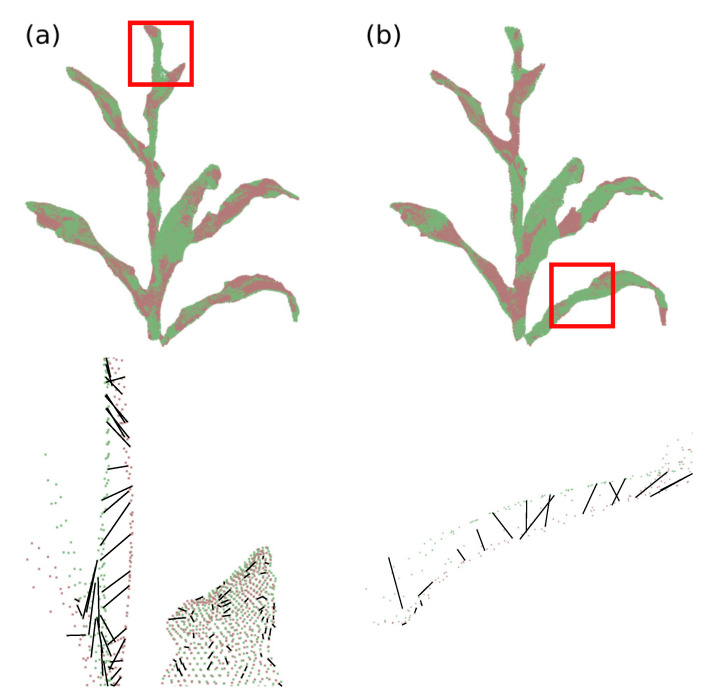
Keypoints correspondence. (**a**) Plant’s stem region, comparison of deformation graphs between frames 30 (green) and 40 (red) and (**b**) plant’s leaf region, comparison of deformation graphs between frames 40 (green) and 50 (red).

**Figure 11 sensors-21-04115-f011:**
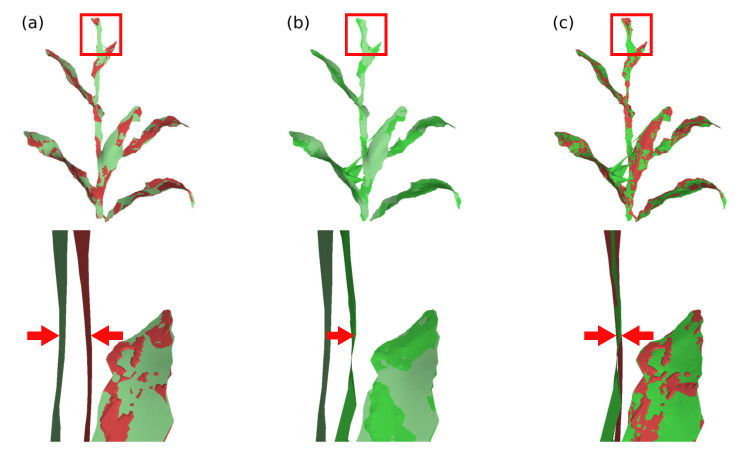
Model deformation in plant’s stem region. (**a**) Difference of the plant’s stem position between the model deformation graph (green) and the input surface deformation graph (red), during processing between frames 30 and 40; (**b**) deformation of the model surface and (**c**) comparison between deformed model surface (green) and input surface (red).

**Figure 12 sensors-21-04115-f012:**
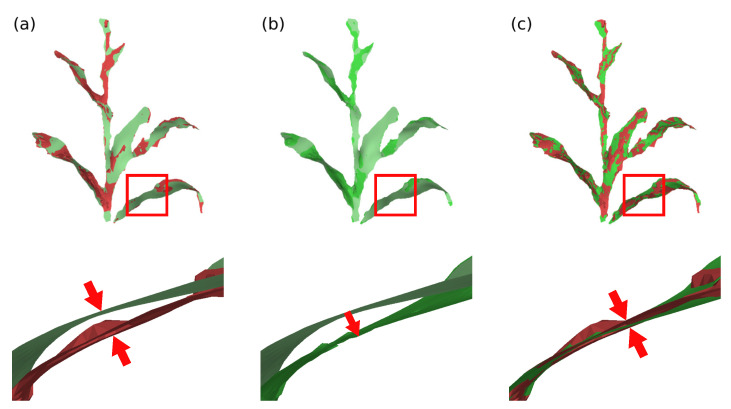
Model deformation in plant’s leaf region. (**a**) Difference of the plant’s leaf position between the model deformation graph (green) and the input surface deformation graph (red), during processing between frames 40 and 50; (**b**) deformation of the model surface and (**c**) comparison between deformed model surface (green) and input surface (red).

**Figure 13 sensors-21-04115-f013:**
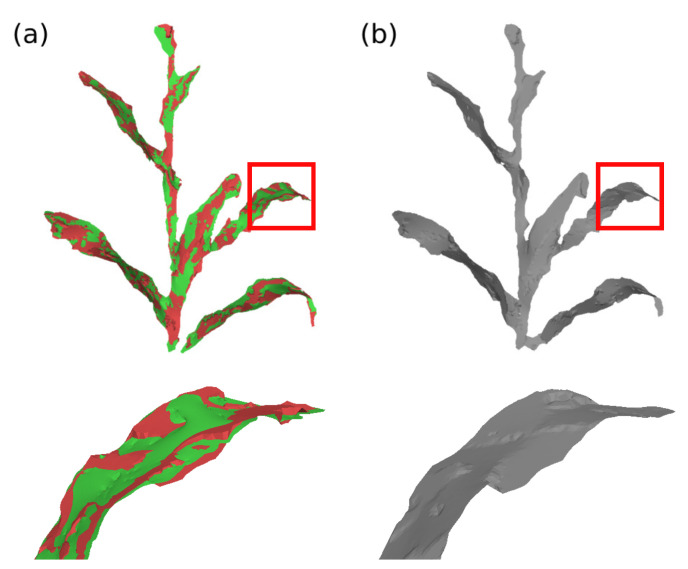
Input and model surfaces fusion. (**a**) Overlap of the deformed model surface (green) and the input surface (red) and (**b**) new model after surfaces fusion.

**Figure 14 sensors-21-04115-f014:**
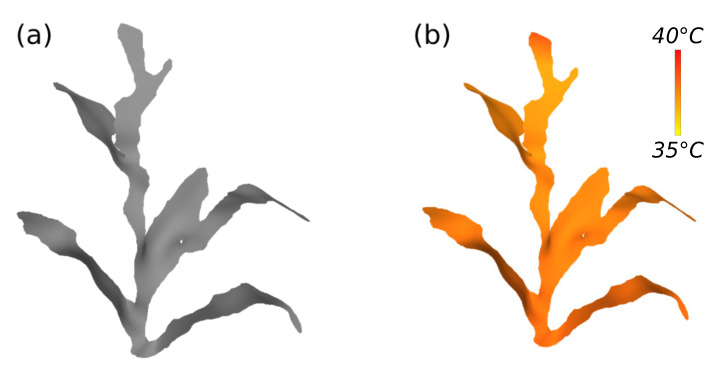
Model and temperature sensor data fusion. (**a**) Model and (**b**) colorized model with data from the temperature sensor.

**Figure 15 sensors-21-04115-f015:**
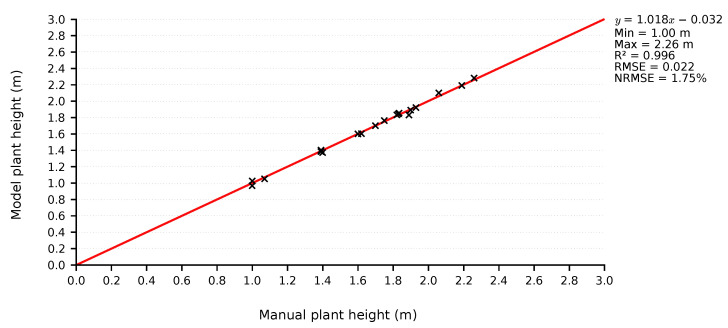
Scatter plot of plant height measurements; comparison between depth-based height and manual height measurements.

**Figure 16 sensors-21-04115-f016:**
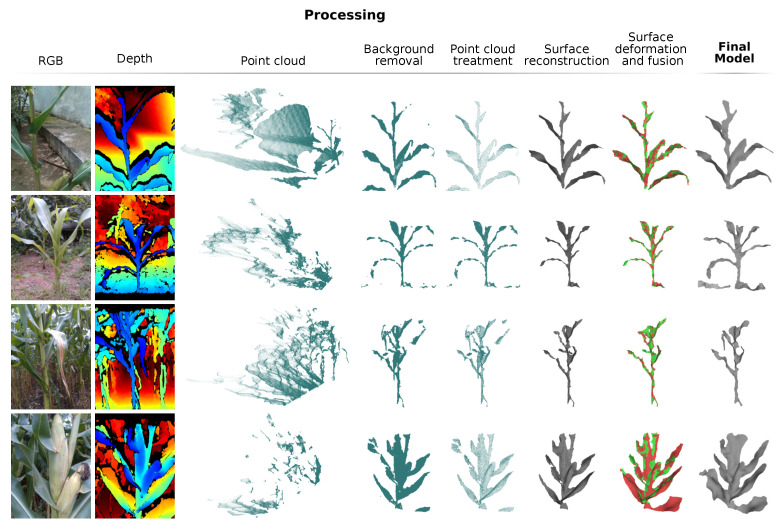
Examples of models generated by the proposed system.

**Figure 17 sensors-21-04115-f017:**
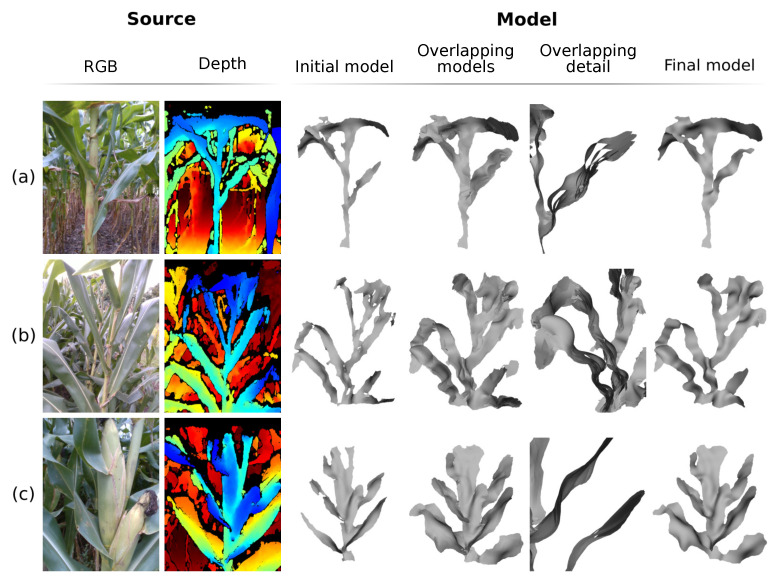
Overlapping of the partial models along the processing for the generation of the plant model. (**a**–**c**) Examples of non-rigid deformation and data accumulation in the generated models.

**Figure 18 sensors-21-04115-f018:**
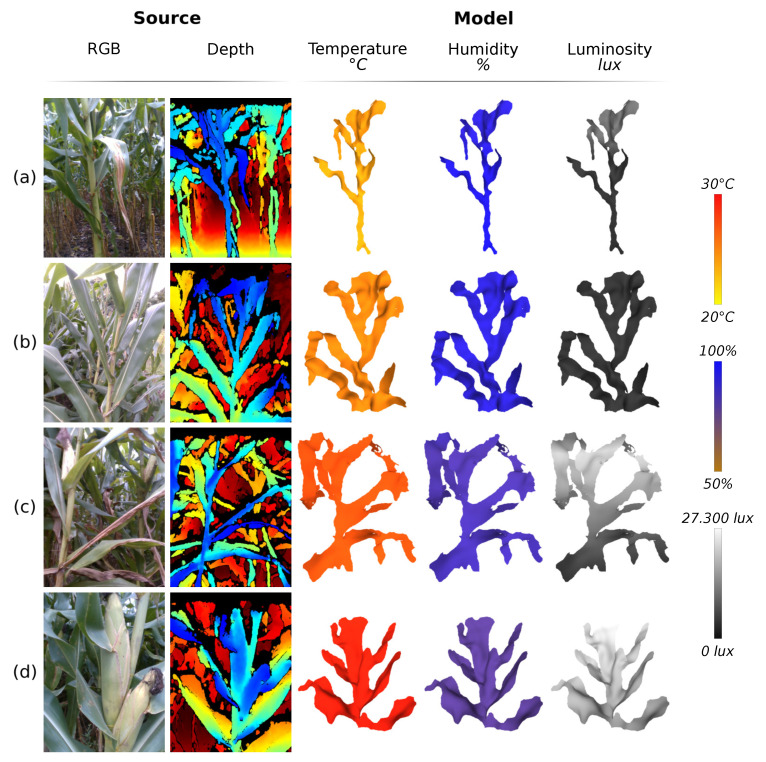
Generated models and fusion with sensor data. (**a**) Plant inside the row with low light incidence (cloudy); (**b**) plant outside the row with low light incidence; (**c**) plant inside the row with high light incidence (sunny); and (**d**) plant outside the row with high light incidence.

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
