# Peer review of "3D Reconstruction of Non-Rigid Plants and Sensor Data Fusion for Agriculture Phenotyping"

_sensors, 2021, doi:10.3390/s21124115_

Round 1
Reviewer 1 Report
The authors investigated the 3D reconstruction of corns by creating a terrestrial acquisition module comprising various sensors, and processing module to clean the data and perform the 3D reconstruction.
The paper is well written and I appreciate the fact that the authors described both modules in details.
Overall the method described is interesting and seem to give satisfying results, although it is sometimes hard to understand the utility of every step of the method and to what extent they really improve the reconstruction process. To this end, an ablation study would greatly help the reader.
As the paper presents a high scientific contribution, with both a robot and the processing pipeline, and with many details on every steps, I accept this paper.
I listed hereafter the remarks that I have regarding this paper:
1- In section 3.2.3, why is it necessary to compute all points to points distance, and why not use an approximation based on voxel to voxel distances? It seems that it would improve performances without deteriorating too much the results
2- The Advancing Front algorithm presented in section 3.2.5 seem similar to the VSA algorithm and I wonder which one would give the best results in this case.
3- Time processing details would be greatly appreciated to see the applicability of the method in large fields.
4- An ablation study would be greatly appreciated to show the utility of each step.
Author Response
In view of the reviewers' recommendations and requests, we present the new version of our manuscript. We have tried to include all points commented by the reviewers in time, as well as balancing requests and comments, especially in section 3.2. We appreciate your attention to our work and we understand that this process seeks to improve its quality. Follow our comments on the questions proposed.
RC: Reviewer Comment
AC: Authors CommentRC: 1- In section 3.2.3, why is it necessary to compute all points to points distance, and why not use an approximation based on voxel to voxel distances? It seems that it would improve performances without deteriorating too much the results.
AC: The outlier removal operation is the last procedure performed to reduce the amount of data for processing by the system. Before this operation, the system performs the point cloud standardization (section 3.2.2), where the cloud points are grouped into voxels according to a predefined resolution. From this grouping, a single point is defined following the rule "centroid of the points belonging to the voxel" to represent the group. Thus, in practice, the system already removes the outliers by the distance between the voxels (or the point defined to represent it). We included the information in the text about the technique used by the system to define the point that represents the group; and we also included in section 5, as a suggestion for improving the system, to analyze other techniques for removing outliers. Please, see the highlighted text (lines 331-333 and 648-650).RC: 2- The Advancing Front algorithm presented in section 3.2.5 seem similar to the VSA algorithm and I wonder which one would give the best results in this case.
AC: We chose the Advancing Front surface reconstruction technique because it is based on the well-known Delaunay triangulation technique and because this technique tries to form the best triangles for the surface definition. Unfortunately, we did not have the opportunity to deep evaluate the VSA (Variational Shape Approximation) technique to compare it with Advancing Front technique applied to the studied dataset. However, we have included in section 5, as a suggestion for improving the system, to analyze other methods for surface reconstruction. Please, see the highlighted text (lines 650-651).RC: 3- Time processing details would be greatly appreciated to see the applicability of the method in large fields.
AC: At this moment we did not develop the system for real-time application, since we developed it for CPU execution. The improvement, in this sense, would be to implement the functionalities of the system using GPU parallelism, which will certainly improve the processing time. Also, the system was design for phenotyping purposes which usually uses small plantation areas. We have tried to describe this in section 5, as we recognize it as an important point for improvement. Please, see the highlighted text (lines 640-646).
RC: 4- An ablation study would be greatly appreciated to show the utility of each step.
AC: In this major revision we focused on presenting for each step the "before" and "after" of the described technique application . We did not perform a deep ablation study, with statistical and performance data for example, but through the figures it became clearer how each technique influences the processing of the point clouds and the generated surfaces. Thus, section 3.2 has been completely revised; we have simplified some explanations and included the figures for comparison. Please, see section 3.2.
Reviewer 2 Report
In this paper, the authors built up a setup to capture the plant's 3D information for agriculture phenotyping. The proposed system was composed of two modules: terrestrial acquisition module and processing module.
- The paper pile up a lot of introductions of the existing methods (Sec3.2.3 outliers removing, Sec3.2.4 bilateral smoothing, Sec3.2.5 seems a copy from [1], Sec.3.2.6 signature of histogram of orientation(SHOT) Sec3.2.7 SR-ARAP) and lack illustration of how to employ these methods to capture the non-rigid corn corps. Also, the authors should add more results (such as figures or tables) of how to utilize these approaches based on captured RGB-D data.
- In Sec 3.1, the authors claim the D435i depth sensor can achieve a good precision (1mm). To my knowledge, the RMS error of the Realsense D435 is only about 2-10mm [2]. The sensor used the near-infrared light to illuminate the speckle pattern, so the sensor's accuracy may decline outdoor (especially on a sunny day). Because the data acquired by the sensor lay the foundation of this paper, the authors should give more proofs.
- In Sec 3.2.1, the author didn't prove the logarithm cutting better than linear cutting. Also, I think the proposed filter is also a type of pass-through filter. Why not use the RGB images for segmentation? Figure 5 needs to be presented in a point cloud instead of depth image.
- Sec 3.2.3-3.2.7 should reduce the introduction of existing methods and add more information on employing these approaches for data processing.
- In the experimental section, the author should add more results (not just RGB image and depth image) to present the whole work.
[1] https://doc.cgal.org/latest/Advancing_front_surface_reconstruction/index.html
[2] https://www.intelrealsense.com/depth-camera-d435i/
Author Response
In view of the reviewers' recommendations and requests, we present the new version of our manuscript. We have tried to include all points commented by the reviewers in time, as well as balancing requests and comments, especially in section 3.2. We appreciate your attention to our work and we understand that this process seeks to improve its quality. Follow our comments on the questions proposed.
RC: Reviewer Comment
AC: Authors Comment
RC: 1- The paper pile up a lot of introductions of the existing methods (Sec3.2.3 outliers removing, Sec3.2.4 bilateral smoothing, Sec3.2.5 seems a copy from [1], Sec.3.2.6 signature of histogram of orientation(SHOT) Sec3.2.7 SR-ARAP) and lack illustration of how to employ these methods to capture the non-rigid corn corps. Also, the authors should add more results (such as figures or tables) of how to utilize these approaches based on captured RGB-D data.
AC: While preparing the article version that was submitted, we tried to describe as best as possible the information about the techniques used in the system, so that the work could be understood by various levels of readers, from students to more experienced researchers. In this new version, we have tried to balance the description of these techniques with the request of the reviewers. Thus, section 3.2 has been completely revised; we simplified some explanations and reduced the information on the techniques described; we also included, for each step, the "before" and "after" by means of figures, making it clearer how each technique influences the processing of the point clouds and surfaces generated. Please, see section 3.2.
RC: 2- In Sec 3.1, the authors claim the D435i depth sensor can achieve a good precision (1mm). To my knowledge, the RMS error of the Realsense D435 is only about 2-10mm [2]. The sensor used the near-infrared light to illuminate the speckle pattern, so the sensor's accuracy may decline outdoor (especially on a sunny day). Because the data acquired by the sensor lay the foundation of this paper, the authors should give more proofs.
AC: In section 3.1 we only presented to the reader some characteristics of the sensors installed in the terrestrial acquisition module. For this reason, we included the accuracy information informed by the camera manufacturer (1mm). Despite this data, we imagine that this accuracy would not be achieved in outdoor environments; thus, we conducted the study presented in section 4 to find out what the real accuracy of the camera would be, an issue that directly influences the accuracy of the model generated. As a result, we found that the RMSE was 0.022 m or, in normalized form, 1.75\% error. These results indicate that the accuracy for the study environment is acceptable and that the camera in question can be used in this type of application. To avoid confusion, we have removed the camera accuracy information (1mm) from section 3.1. Please, see the highlighted text (lines 584-593) and figure 15.
RC: 3- In Sec 3.2.1, the author didn't prove the logarithm cutting better than linear cutting. Also, I think the proposed filter is also a type of pass-through filter. Why not use the RGB images for segmentation? Figure 5 needs to be presented in a point cloud instead of depth image.
AC: The first operation performed by the processing system, after reading the files with the collected data was the background removal. Indeed, the RGB images were an option to perform this operation, however two factors were found that made the segmentation of these images difficult. The first one was the color pattern; in the images collected from the plantation the RGB image is basically composed of shades of green, making color segmentation difficult. The second factor was the pattern of the shapes in the image, making segmentation by shapes difficult. So the best way we found to remove the background was to use the depth data and prioritize the points closest to the camera. Indeed, the proposed technique is a "pass-through filter", but the way of defining the background distance value is different from the linear way, based on the logarithmic function. By the characteristic of this function, we realized that it would be possible to develop a way to calculate the background distance that prioritizes the elements closer to the camera and is less sensitive to the variation of minimum and maximum distances captured by the camera. Figure 4 exemplifies how efficient the proposed technique is in varying these distances, since the background varies less with the change of distance compared to the removal by the linear function. Furthermore, it is possible to control this sensitivity by means of the logarithmic base, making it more flexible than the linear removal. We include more details about these features in section 3.2.1. Figure 5 has been updated. Please, see the highlighted text (lines 265-274 and 300-305).
RC: 4- Sec 3.2.3-3.2.7 should reduce the introduction of existing methods and add more information on employing these approaches for data processing.
AC: We have performed the requested reductions and included figures to make it more understandable what each technique accomplishes during model generation. Please, see section 3.2.
RC: 5- In the experimental section, the author should add more results (not just RGB image and depth image) to present the whole work.
AC: We followed the recommendation and updated the figure (which is now figure 16), to better exemplify the processing steps for generating the model.
Reviewer 3 Report
The paper presents an approach for the 3D reconstruction of non-rigid plants and sensor data fusion for agriculture phenotyping. The paper is interesting and well written. The method and the results are sound, properly described and analyzed. The following points need to be considered to improve the quality of the manuscript.
1) The main concerns is that the main contributions of the present study are not clearly stated with respect to the state-of-the-art in the field.
2) The data acquisition system should be better described and commented. The frequency rate of the different sensors should be added to the manuscript.
3) I do not understand why the authors mention an aerial subsystem in the abstract if this one is actually not considered in the paper.
4) The following references are suggested to improve the literature review:
Ristorto, G., Gallo, R., Gasparetto, A., Scalera, L., Vidoni, R., & Mazzetto, F. (2017). A mobile laboratory for orchard health status monitoring in precision farming. Chemical engineering transactions, 58, 661-666.
Wang, L., Xiang, L., Tang, L., & Jiang, H. (2021). A Convolutional Neural Network-Based Method for Corn Stand Counting in the Field. Sensors, 21(2), 507.
Oliveira, L. F., Moreira, A. P., & Silva, M. F. (2021). Advances in Agriculture Robotics: A State-of-the-Art Review and Challenges Ahead. Robotics, 10(2), 52.
Author Response
In view of the reviewers' recommendations and requests, we present the new version of our manuscript. We have tried to include all points commented by the reviewers in time, as well as balancing requests and comments, especially in section 3.2. We appreciate your attention to our work and we understand that this process seeks to improve its quality. Follow our comments on the questions proposed.
RC: Reviewer Comment
AC: Authors Comment
RC: 1) The main concerns is that the main contributions of the present study are not clearly stated with respect to the state-of-the-art in the field.
AC: During the literature review process, especially in the analysis of works on 3D reconstruction of plants, we realized where our work could be developed. There are works that perform this task in controlled environments and others that propose systems for outdoor environments. Despite promising results, we did not find any work that performed the 3D reconstruction of plants in a non-rigid way; so we propose a system, which we believe is for the first time, that performs this kind of reconstruction for agricultural purposes. This is a point that we really need to make clear. So we made some adjustments to emphasize the importance of the work and position it in the state-of-the-art of the field. (lines 4; 40-58; 65-74; 154-160 and 669-674).
RC: 2) The data acquisition system should be better described and commented. The frequency rate of the different sensors should be added to the manuscript.
AC: Section 3.1 deals with the description of the terrestrial acquisition module. In this section we describe the main aspects of the acquisition system such as: equipment, with details of the robot, microcontrollers, sensors and camera; architecture for control and integration of the components (ROS); communication protocols used ($\mathrm{I^{2}C}$, Serial); and materials for assembling the cabling. We include the measurement time information for the sensors and make clear the frequency of data collection. In addition, we comment that other configurations for data collection are possible. Please, see the highlighted text (lines 211-216 and 218-222).
RC: 3) I do not understand why the authors mention an aerial subsystem in the abstract if this one is actually not considered in the paper.
AC: The system presented in this paper is part of a larger system, where other data is collected by drones (RGB-D and thermal images). We wanted to introduce this idea, because in the future complementary works can be developed with these data. But in order not to cause confusion we removed this information from the abstract, but we kept it in the introduction (section 1) and in the discussion (section 5).
RC: 4) The following references are suggested to improve the literature review:
Ristorto, G., Gallo, R., Gasparetto, A., Scalera, L., Vidoni, R., Mazzetto, F. (2017). A mobile laboratory for orchard health status monitoring in precision farming. Chemical engineering transactions, 58, 661-666.
Wang, L., Xiang, L., Tang, L., Jiang, H. (2021). A Convolutional Neural Network-Based Method for Corn Stand Counting in the Field. Sensors, 21(2), 507.
Oliveira, L. F., Moreira, A. P., Silva, M. F. (2021). Advances in Agriculture Robotics: A State-of-the-Art Review and Challenges Ahead. Robotics, 10(2), 52.
AC: We followed the recommendation and reviewed the articles indicated. Two of them were included in our references:
REF 20:
Ristorto, G., Gallo, R., Gasparetto, A., Scalera, L., Vidoni, R., Mazzetto, F. (2017). A mobile
laboratory for orchard health status monitoring in precision farming. Chemical engineering
transactions, 58, 661-666.
REF 33:
Oliveira, L. F., Moreira, A. P., Silva, M. F. (2021). Advances in Agriculture Robotics: A State-
of-the-Art Review and Challenges Ahead. Robotics, 10(2), 52.
The third article, despite performing activities in agriculture, does not use robotic equipment and does not deal with 3D reconstruction.
Round 2
Reviewer 2 Report
The submission has been improved after revision. However, this paper mainly describes a system for plant 3D reconstruciton, and similar works have also been reported before. The overall technical novelty is quite limited.
Author Response
First of all, we would like to thank you for your attention to our work; in the first round of revisions, we have tried our best to follow the reviewers' recommendations to improve the paper quality. In addition to meet the reviewers' requests, we have reviewed it to ensure that the paper's contributions are presented in an appropriate manner.
RC: Reviewer Comment
AC: Authors Comment
RC: The submission has been improved after revision. However, this paper mainly describes a system for plant 3D reconstruction, and similar works have also been reported before. The overall technical novelty is quite limited.
AC: In the literature review we did not find any previous work proposing a system to performs the 3D reconstruction of non-rigid plants in outdoor environments. Thus, we studied a set of techniques and selected the ones with the best performance to present a system capable of performing 3D reconstruction of plants in uncontrolled outdoor environments. In addition to these contributions, we present a terrestrial acquisition module, to collect the data in the field; and novel techniques for the background removal and for the fusion of the data collected by the sensors with the models generated. The great benefit of the proposed system is to allow the reconstruction of the plants considering their movement, caused by wind, for example; and to allow the data to be collected in the field dynamically (videos) and not only in greenhouses or statically (photos).
Thus, in summary, the work presents the following innovations:
- Novel system to perform 3D reconstruction of plants in uncontrolled outdoor environments, using a set of techniques, with proven efficiency in the literature.
- Novel technique to perform background removal from point clouds, based on the logarithmic function.
- Novel way to fusion sensor data (temperature, humidity and luminosity) with the generated models, using linear interpolation to colorize the models in a gradient fashion.
We address the contributions in several parts of the paper. Please, see the highlighted text (lines 4; 40-58; 65-74; 154-160 and 669-674).
Reviewer 3 Report
The paper was improved with respect to the previous version. I suggest the paper to be accepted for publication.
Author Response
Thank you for your attention to our work; in the first round of revisions, we have tried our best to follow the reviewers' recommendations to improve the paper quality.